# Intermittent hypoxia in neonatal rodents affects facial bone growth

**Eung-Kwon Pae**[1]***, Ronald M. Harper**[2]

**1** Department of Orthodontics and Pediatric Dentistry, School of Dentistry, University of Maryland, Baltimore, MA, United States of America, **2** Department of Neurobiology, David Geffen School of Medicine, University of California at Los Angeles, Los Angeles, CA, United States of America

* epae@umaryland.edu

**Data Availability Statement:** All relevant data are within the paper and its Supporting Information files.

**Funding:** EP received Biomedical Research Award from the American Association of Orthodontists Foundation (AAOF) at https://www.aaofoundation.

## Abstract

Preterm human infants often show periodic breathing (PB) or apnea of prematurity (AOP), breathing patterns which are accompanied by intermittent hypoxia (IH). We examined cause-effect relationships between transient IH and reduced facial bone growth using a rat model. Neonatal pups from 14 timed pregnant Sprague-Dawley rats were randomly assigned to an IH condition, with oxygen altering between 10% and 21% every 4 min for 1 h immediately after birth, or to a litter-matched control group. The IH pups were compared with their age- and sex-matched control groups in body weight (WT), size of facial bones and nor-epinephrine (NE) levels in blood at 3, 4, and 5-weeks. Markedly increased activity of osteoclasts in sub-condylar regions of 3-week-old IH-treated animals appeared, as well as increased numbers of sympathetic nerve endings in the same region of tissue sections. Male IH-pups showed significantly higher levels of NE levels in sera at 3, 4 as well as 5-week-old time points. NE levels in 4- and-5-week-old female pups did not differ significantly. Intercondylar Width, Mandible Length and Intermolar Width measures consistently declined after IH insults in 3- and 4-week-old male as well as female animals. Three-week-old male IH-pups only showed a significantly reduced (p < 0.05) body weight compared to those of 3-week controls. However, female IH-pups were heavier than age-matched controls at all 3 time-points. Trabecular bone configuration, size of facial bones, and metabolism are disturbed after an IH challenge 1 h immediately after birth. The findings raise the possibility that IH, introduced by breathing patterns such as PB or AOP, induce significantly impaired bone development and metabolic changes in human newborns. The enhanced NE outflow from IH exposure may serve a major role in deficient bone growth, and may affect bone and other tissue influenced by that elevation.

## Introduction

We previously reported that exposure to intermittent hypoxia (IH) immediately after birth in a rat model results in deficient bone development in peripheral limbs [1] raising a concern that similar breathing patterns in human infants may comparably impair osteogenesis in other structures, particularly those of facial bones. The concern is significant; premature infants

net/ for the study. PI was a sole investigator. This award supported the study fully. The funders had no role in study design, data collection and analysis, decision to publish, or preparation of the manuscript.

**Competing interests:** NO authors have competing interests.

often exhibit periodic breathing (PB) or apnea of prematurity (AOP) [2, 3], which results in IH exposure, often to worrying levels, and birthrates of preterm infants (under 37 weeks gestation) represent approximately 10% of live births for the last two decades in the US [4, 5] and during the COVID-era [6]. PB and AOP disrupt autonomic nervous system (ANS) activity [7–9], with increased sympathetic tone [10] in preterm infants, affecting autonomic measures such as blood pressure, heart rate, and endocrine release, and potentially compromising survival [11, 12]. PB or AOP elevates postganglionic norepinephrine (NE) outflow [13–15], leading to increased NE levels in blood. Elevated serum NE levels from sustained sympathetic outflow decreases bone formation and increases bone resorption *via* NE coupling with β2 adrenergic receptors that are abundant in bones [16–18]. NE secretion from the adrenal glands increases as stress levels remain high *via* IH disruptions to the ANS. The elevated NE binds to β2 adrenergic and other adrenergic receptors in osteoblasts in bones, including orofacial bones. Presumably, levels of NE in blood are proportional to the levels of sympathetic outflow [19, 20].

Newborn rat pups have been used as a viable model mimicking preterm human infants for delayed neural development; rat pups at postnatal day 0 (P0) are considered equivalent to neural development of human neonates at gestational week 25 [21]. Thus, we assume exposing P0 rat pups to IH parallels IH effects accompanying PB that are frequently observed in human infants born preterm. A harmful relationship between PB (inducing IH) and disturbed bone development was recognized only recently [22, 23], although PB traditionally has been considered to be relatively harmless in the pediatric community [24].

We reported markedly diminished bone hardness in mandibular basal bones and hind limbs 3 weeks after IH-insults at P0 [1]. In the current study, we further investigated orofacial bone quality and body weight in relation to IH exposure and to increased levels of NE in blood based on the h*ypothesis*: *A brief intermittent hypoxic challenge decreases mandibular bone quality, size of maxilla and mandibles, alters body weight in rat pups, and is accompanied by elevated norepinephrine levels.* We investigated potential cause-effect relationships between IH and a lack of mineral density in orofacial bones to understand several clinical studies reporting significant osteopenia in infants and developing children with a history of premature birth [22, 23].

## Materials and methods

### Experimental design

As previously described [1], near end-term pregnant Sprague Dawley rats (n = 14) were acquired from Jackson Labs and maintained until parturition. In less than 12 hours after birth, offspring of the experimental group were housed for 1 h within a temperature controlled hypoxic chamber (Billups-Rothenberg Inc., San Diego, CA, USA), in which $O_2$ concentration alternated between room air (approximately 21% $O_2$) and hypoxic conditions (approximately 10% $O_2$) every 4 min (Fig 1). We treated the pups from randomly-selected 7 mothers for 1 h because we found that 1 h treatment (or 7 cycles of IH exposure) reflects a clinically not unusual as well as sufficiently mild condition devoid of conspicuous neural damage. Randomly-selected offspring and 7 mothers of the control group were exposed to ambient air. Litter size was matched between the control and IH groups to help equalize weights of pups. After IH treatment, pups were maintained in ambient air until euthanasia under the same conditions as the control group. A total of 136 pups were euthanized when they were 3, 4 and 5 weeks old. All processes involved in the experiments followed the National Institutes of Health guide for the care and use of Laboratory animals as approved (IACUC #D121101, University of Maryland, Baltimore).

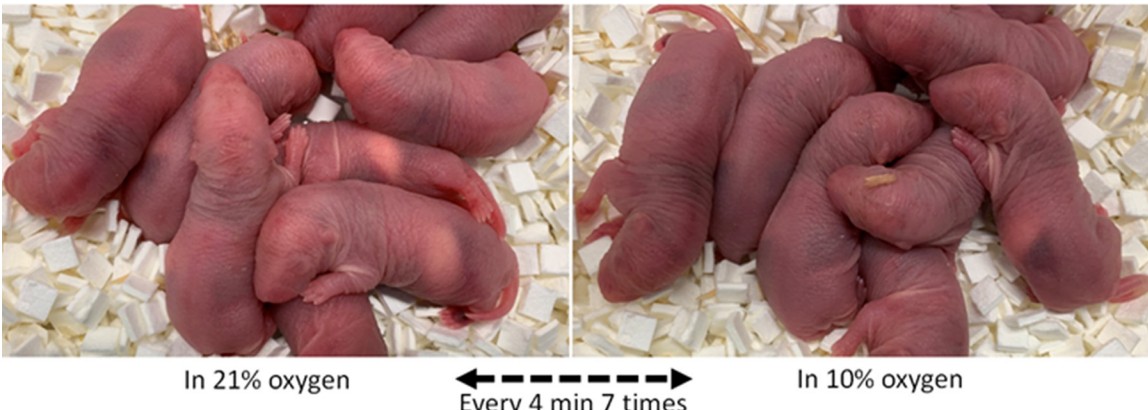

In 21% oxygen

← – – – →
Every 4 min 7 times

In 10% oxygen

**Fig 1. Control and IH-exposed neonatal Sprague-Dawley rat pups less than 12 hours of age.** Note that skin color of the pups under the hypoxic condition has a bluish tinge compared to that of control pups. Temperature of the ambient air for both control and experimental animals was controlled at approximately 25 degrees Celsius.

## Euthanasia and sample procurement

Experimental and control pups were fasted for 2 h prior to euthanasia. For tissue procurement, the animals were sacrificed in a $CO_2$ chamber, and blood was drawn from the left ventricle of the heart immediately. Decapitated heads were rapidly harvested from the animals and stored at −20˚C. Blood and tissue samples were harvested, and serum was separated from centrifuged blood immediately and secured for assays. For images by immunohistochemistry staining, fresh bone samples were fixed in 10% formalin overnight and washed with tap water for several minutes, then decalcified in Leica Decalcifier II (LeicaBiosystem, Cat#: 3800421) for 8 hours. The tissue samples were embedded in paraffin and sectioned at 4μm thickness, baked at 60 degrees for 3–4 hours. For histochemistry staining, samples were fixed in 10% NBF (Neutral Buffered Formalin) for 24 hours at room temperature.

## ELISA assay

Assays for norepinephrine (NE) quantification from serum were performed using an ELISA Kit (LS-F10598-1, LS Bio, WA, USA), in accordance with the manufacturer's protocol. Shortly, samples in an equal amount were incubated in NE monoclonal antibody-coated plates, detected using a secondary biotinylated antibody, and followed by Avidin-Horseradish Peroxidase (HRP) complex. Unbound HRP-conjugate was washed away. Immobilized antibody–enzyme conjugates were quantified by monitoring color development in the presence of TMB (3,3′,5,5′-tetramethylbenzidine) substrate at 450 nm in a microplate reader (SpectraMax® iD3, Molecular Devices, San Jose, CA).

## Immunohistochemistry

Paraffin on the selected slides was removed with xylene and rehydrated through graded ethanol. Heat-induced antigen retrieval (HIER) was carried out for sections in 0.001 M EDTA buffer, pH = 8.00 using a Biocare decloaker at 60 degree for 18 hours. The slides were then stained with Tyrosine Hydroxylase (TH) antibody (Millipore, Cat#: ab1542, 1–200) for 1 h at room temperature, the section incubated with Rabbit anti-Sheep secondary antibody (Thermofisher, Cat#: PI31240, 1–500), and the signal was detected using Leica Refine RED Kit (LeicaBiosystem, Cat#: DS9390) on Bond RX auto-Stainer.

### H & E / tartrate-resistant acid phosphatase (TRAP) staining

H&E and tartrate-resistant acid phosphatase (TRAP) staining were performed according to the manufacturer's protocols (Sigma, St. Louis, MO, United States) as described earlier [25]. Stained sections were scanned and analyzed using the Aperio Scanscope CS instrument (Aperio Scanscope CS system, Vista, CA, United States). The number of TRAP-positive osteoclasts and cuboidal osteoblasts adherent to the bone surface in the region of interest were quantified using Fiji (ImageJ) software.

### Total body weights and linear measurements on facial bones

Total body weight was measured by a scale (Denver Instrument XE-510) at 0.01 g accuracy immediately after death of each animal. Frozen decapitated heads were outsourced to be defleshed by dung-beetles (Skinner's Skull Shop, Bedford, WY 83112). A digital caliper (Whitworth, 0-150mm METR-ISO) was used to measure linear variables on maxilla and mandibles, as shown in Fig 2.

### Statistics

Data were analyzed and graphed using the statistical software GraphPad PRIZM 7 (GraphPad Software, La Jolla, CA) and SPSS v.23 (IBM). All statistical inference tests were two-tailed;

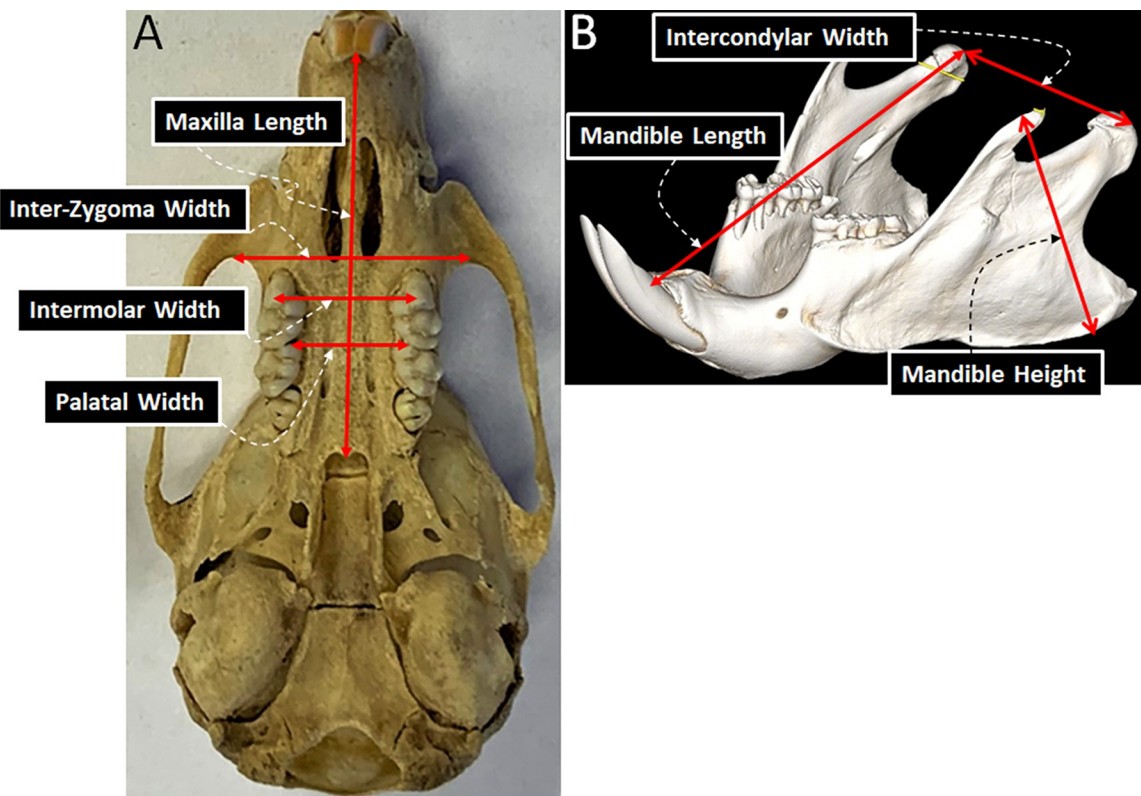

**Fig 2. Linear measurements estimate of the size of the maxilla and the mandible of rats.** A. Transverse view of the maxilla and the skull base: Intermolar width–distance between upper buccal cusp tips of the maxillary first molars; Palatal width–linear distance between the right and left interdental alveolar bone of the first and the second upper molars; Inter-Zygoma width–distance between the most posterior curvature of the zygoma; Maxilla length–distance between the middle of interdental alveolar bones of the upper central teeth and posterior nasal spine; B. Sagittal view of the mandible from approximately 45 degrees anterior and superior: Intercondylar width–distance connecting the midpoint of the right and left condylar heads; Mandible length–distance between the midpoint of the condylar head and interdental bone of the lower central incisors; Mandible height–longest distance between the lower and upper border of the ramus.

p < 0.05 was considered significant. For group comparisons between subgroups (control *vs.* IH as well as 3 *vs.* 4 *vs.* 5 weeks), unpaired t-tests (or Mann-Whitney test) and one-way ANOVA were performed. Tissue samples from at least three animals or three independent cultures were measured 3 times for each ELISA assay. Studies on random errors were performed on 4 linear measurements (Intermolar width, Inter-Zygoma width, Mandible length, Intercondylar width) using Dahlberg's formula.

## Results

### Measurement errors

An error measurement study was performed to evaluate intra-examiner reliability. The Dahlberg error D was calculated using $D^2 = \sum_{i=1}^{N} di/2N$, where d$i$ is the difference between the first and second measure; N is the sample size. Based on each Dahlberg error D shown in Table 1, we concluded that the extent of random error was smaller than 0.25 mm; thus, 'small enough' for this study [26].

### Comparisons of NE levels in blood between IH pups vs. control pups

Results of ELISA assays showed that NE levels in blood remained consistently higher in IH-male groups at postnatal 3, 4 and 5 weeks compared to those of age-matched control male pups (Fig 3). However, female pups showed significantly higher NE levels after IH at 3 weeks of age only. The differences in NE measurements between control and IH animals disappeared in 4-week and 5-week- old female pups.

### Weight differences: IH treated group vs. control group

**Male pups.**   Differences in body weight (WT) were compared between IH and control male pups. We found that body weights differed significantly (p < 0.05) 3 weeks after one-time 1h IH exposure of pups at P0. A significant difference of 20% of mean body weights (7.4 g) emerged, as shown in Fig 4. In 4-week-old groups, the weight difference between control (n = 8) and IH (n = 8) pups disappeared (75.4 ± 2.44 g for control group vs. 78.36 ± 2.52 g for IH group). When animals were 5 weeks of age, IH pups (n = 8) were significantly heavier than control pups (n = 5) p = 0.0001 (102.1 ± 1.52 g for control group *vs.* 116.4 ± 1.64 g for IH group).

**Female pups.**   IH-treated female pups were heavier than control female pups in all subgroups. When control pups (n = 5) were compared to IH pups (n = 5) in 3-week-old females, the IH-exposed pups (57.1 ± 0.84 g) were significantly heavier than the control pups (45.6 ± 0.62 g) at p < 0.0001. In 4-week-old animals, IH pups (n = 4) were heavier than control pups (n = 8), (66.8 ± 3.45 g *vs.* 70.73 ± 4.54 g, p = 0.52). When control female pups (n = 8) were compared to IH female pups (n = 3) in the 5-week-old subgroup, IH pups (106.2 ± 3.18 g) were significantly heavier than control pups (92.06 ± 1.82) at p = 0.003.

**Table 1. Dahlberg D values for measurement errors calculated on 5 skulls.**

|  | Intermolar Width | Inter-Zygoma Width | Mandible Length | Intercondylar Width |
|---|---|---|---|---|
| $\sum_{i=5}^{N} di$ | 0.18 | 0.53 | 0.61 | 0.57 |
| **D** | 0.13 | 0.23 | 0.25 | 0.24 |

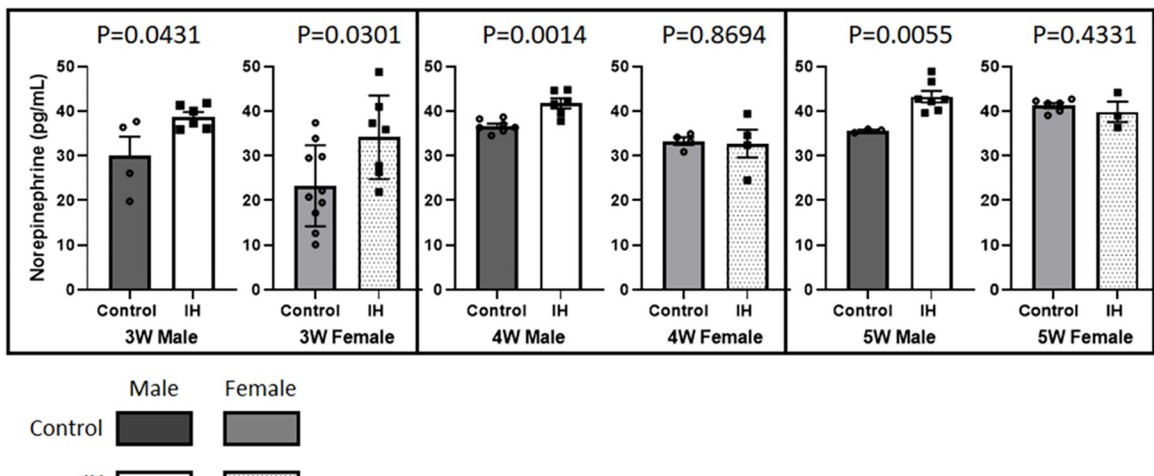

**Fig 3. Comparisons between control vs. IH on the levels of NE in blood.** NE levels in 3-week-old pups (30.0 ± 4.27 vs. 38.7 ± 1.07 for males and 23.3 ± 2.86 vs. 34.1 ± 2.54 for females); 4-week-old pups (36.7 ± 0.54 vs. 41.7 ± 1.12 for males and 33.2 ± 0.86 vs. 32.7 ± 3.10 for females); 5-week-old pups (35.6 ± 0.24 vs. 43.2 ± 1.27 for males and 41.2 ± 0.57 vs. 39.8 ± 2.29 for females). NE levels in blood tended to increase gradually with age, as shown particularly in control female pups (23.3 → 33.2 → 41.2 pg/mL).

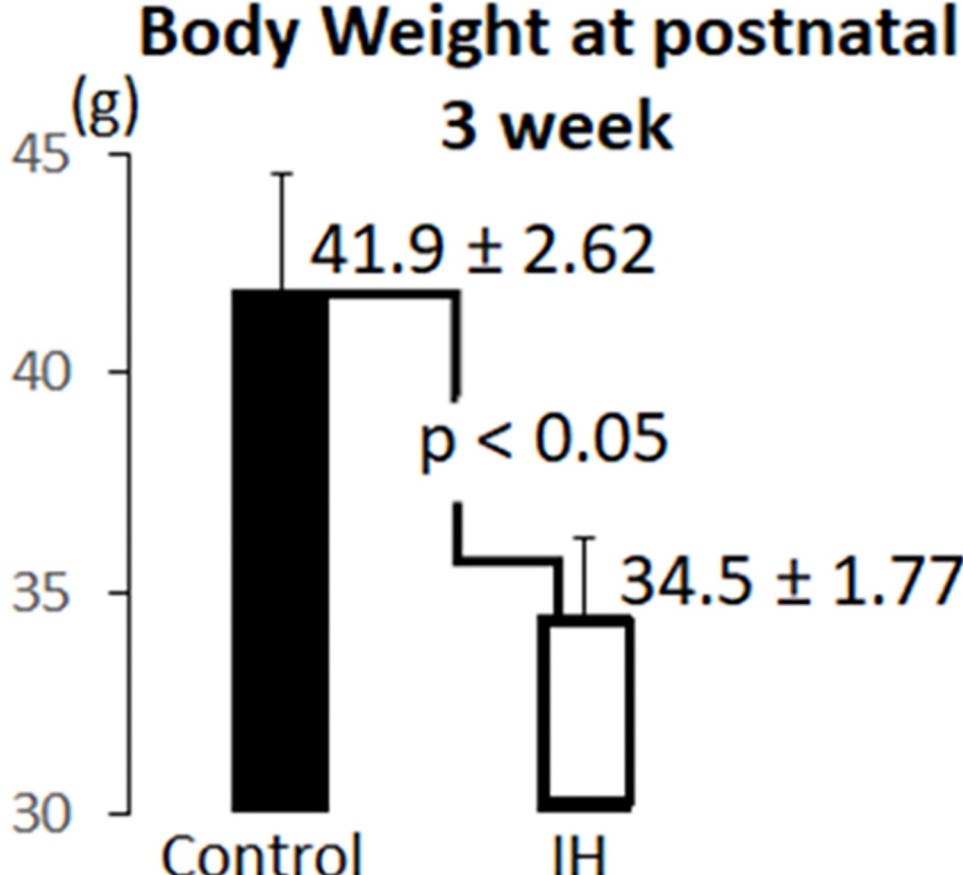

**Fig 4. Body weight measurements on control (n = 7) and IH (n = 3) males were compared.** A catch-up growth in body weights was apparent at the 4th and 5th week time points.

## Comparisons of weights of the dry mandible: IH treated group vs. Control group

Weights of dry mandible were compared between control and IH pups to estimate the portion of bone weights only. As shown in Fig 5, IH effects on dry mandible weight only affected dry weight at 5 weeks (Fig 5).

## Histological differences in sub-condylar bones

Osteochondral interface areas in mandibular condyles were examined, since those areas demonstrate a constant ossification process during the active growth period between P0 and P35.

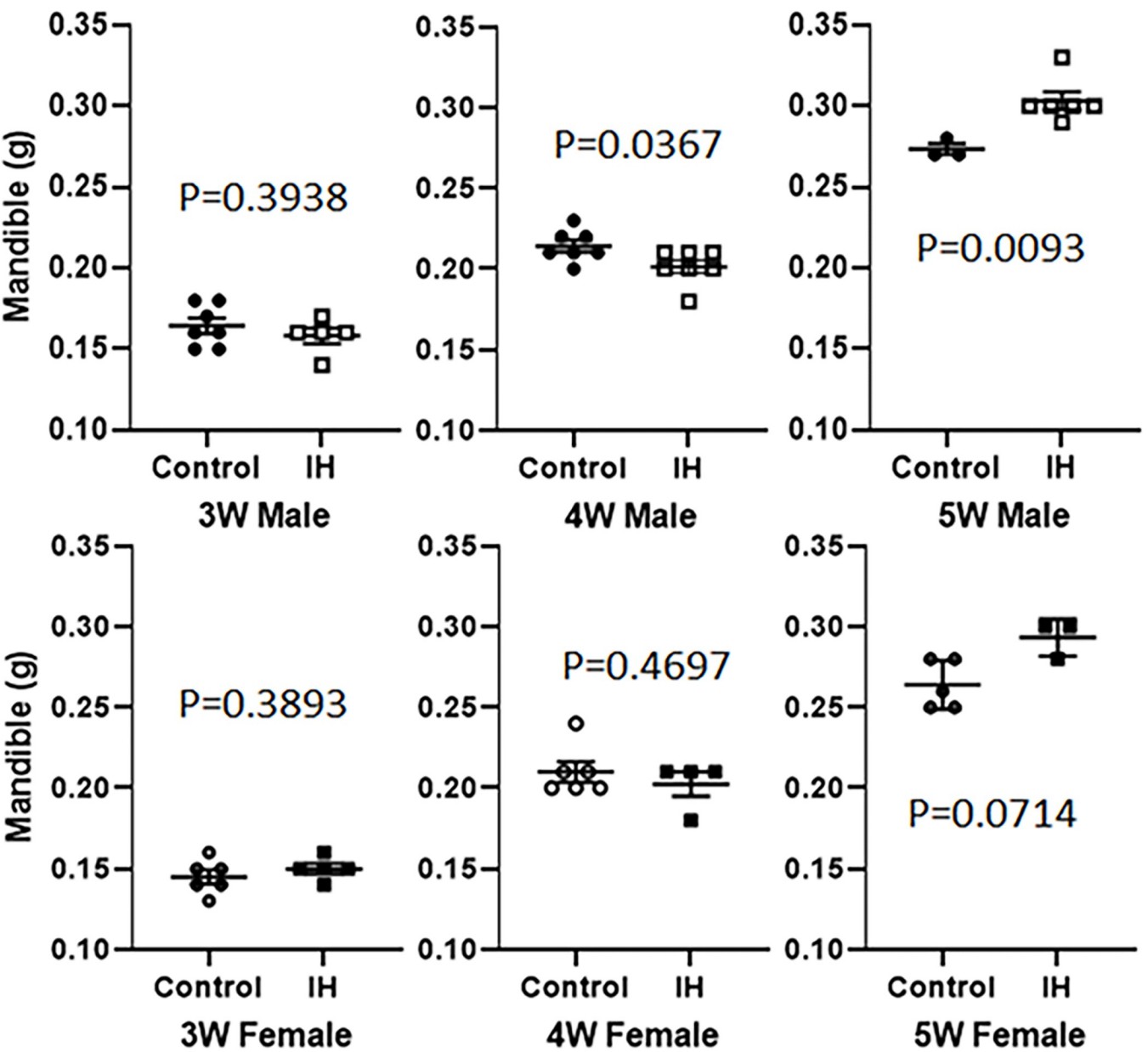

**Fig 5. Weights of dry mandibles that show a significant difference between control and IH animals at 5 weeks.**

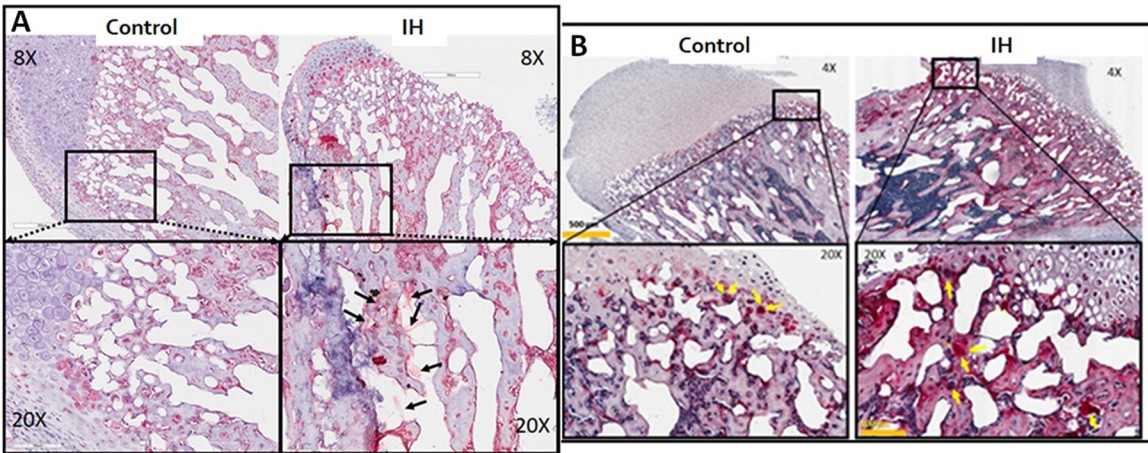

**Fig 6. Differences in histology of the mandible between control vs. IH pups.** A. Immunohistochemical stain using tyrosine hydroxylase (TH) antibody. Sprouting TH-positive sympathetic nerve fibers indicated by arrows in the 20X view of bone section obtained from an IH-treated pup are evident. The area exhibits the osteochondral interface of an IH-treated mandible condyle. B. Condylar heads harvested from control and IH-treated male pups. IH condyle shows an increased TRAP activity with more osteoclasts (indicated by yellow arrows) in the region of interest. Scale bars in 20X panel = 100 μm. Scale bars in 4X panel = 500 μm.

We observed sprouting sympathetic nerve fibers indicated by tyrosine hydroxylase (TH) staining in this subchondral area of 35 days-old IH treated condyles, but no such activity was noted in the same area of the age matched control condyles (See Fig 6A). The differences between the groups were visually evident; thus, we did not count the number of fibers. We compared the osteochondral areas again for osteoclastic activities using a tartrate-resistant acid phosphatase (TRAP) stain. IH condyles show an increased TRAP activity in red color. Significantly more numbers of TRAP-stained osteoclasts in red color were observed in the condyles harvested from IH pups. Thus, we did not perform statistical analysis on the apparent results. Altogether, osteoclastic activity in sub-condylar bones appears to be substantially increased in IH-treated animals.

## IH effects on morphometry of the maxilla and the mandible

IH effects on the size of facial bones were estimated using a total of 6 linear variables in growing pups. We dropped Maxillary length measurements from the following figure presentations because none of the Control vs. IH comparisons of the Maxillary length yielded a statistically significant difference. When control male pups were compared to IH-treated male pups, the IH influences on facial bone growth were evident in 3- and 4-week-old animals. In contrast, 5-week-old pups in the IH groups showed longer length measurements of the maxilla and mandible (Fig 7). A transverse defect in the maxilla *i.e.*, a reduction of intermolar widths in males and females was a primary outcome in 3- and 4-week-old pups.

**Male pups.** Among the measurements, IH-effects on Intercondylar width were noted in 3 week-old pups. IH-effects on mandible growth were found in 4-week-old animals at peak in Intercondylar width, Mandible length and Mandible height measurements (Fig 7). The IH-treated mandibles were shorter (Mandible Length) and smaller (Mandible Height) as well as narrower in the Intercondylar width measurements compared to those of 4 week-old control male pups. Yet, when they become 5 weeks-old, the pups appeared to start a 'catchup growth' in maxilla and mandible bones, *i.e.* IH effects shown at young ages disappeared.

**Female pups.** Five-week-old female pups also exhibited significant growth in maxilla (Palatal width) and mandible (Length) bones; thus, IH pups showed larger maxilla and mandibles

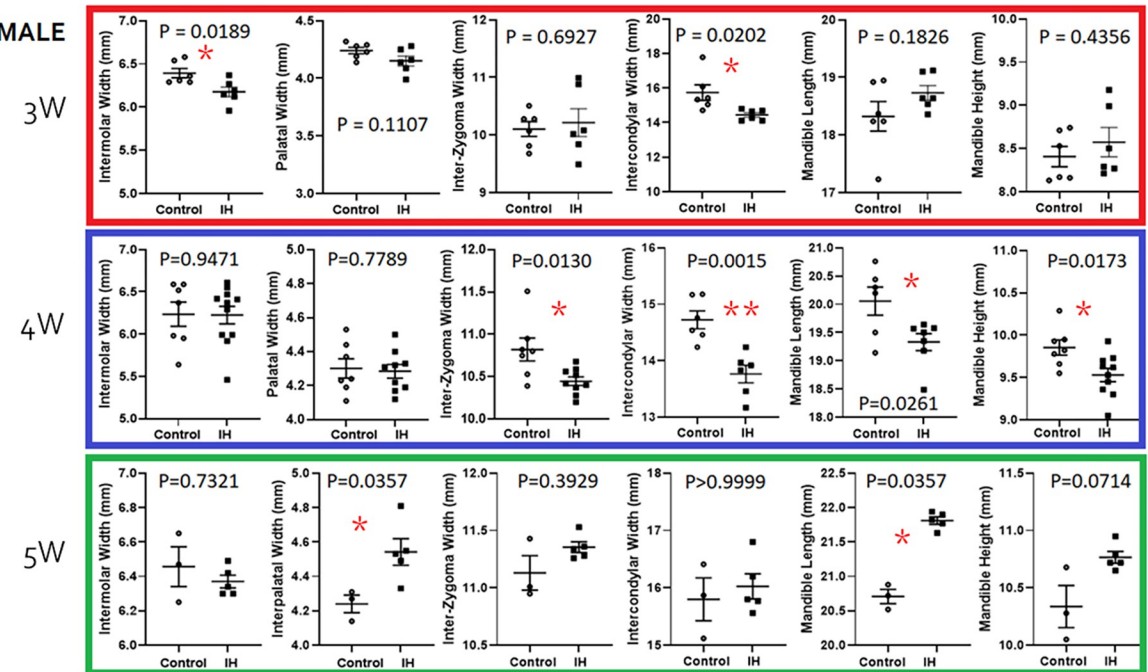

**Fig 7. Morphometric comparisons between the control vs. IH groups in 3 (red), 4 (blue) and 5 (green) week old male pups.**
Although means and standard errors are denoted as shown in each graph, non-parametric methods were used to test implications for small sample size. * < 0.05, ** < 0.01.

at week-5. Yet, 3-week-old pups showed a narrow Intermolar Width, Inter-Zygoma Width, and Intercondylar width (Fig 8).

## Discussion

Many preterm infants who survive early life difficulties face a risk of lifelong health consequences, including abnormal glucose regulation, slow body weight increase, high blood pressure, increased risk of heart failure [27–29], and poorer sexual life [30]. Among these various morbidities, increased bone fragility [31, 32], including a high incidence of facial bone fractures in the course of traffic accidents [33] and a more prevalent metabolic problems [34] among patients born preterm may lead to significant adverse effects of interest to developmental clinicians.

Our hypothesis was that a brief exposure to IH in neonatal rat pups elevates blood levels of NE resulting in a disturbance in facial bone development and metabolic issues related to weight gain. If this hypothesis is supported, further clinical studies should follow, since the annual rates of preterm birth remain high, approximately 10% or higher in the US [6, 35]. Even though the mortality rate of premature infants has improved from advances in neonatal-perinatal medicine [13], post-natal survival is a concern. For instance, the chances of survival at 24 to 25 gestational weeks range from 75% to 90% [36]. The increased potential prevalence of facial bone defects is a challenge to the dental profession. Outcomes of several previous European studies suggest that individuals who experienced preterm birth tend to show malocclusions due to disharmonious facial growth [37]. Although controversial, many previous malocclusion studies were based on the assumption that life-supporting equipment maintained in the oral cavity of preterm infants is the principal causal contributor to malocclusions [38, 39]. However, the findings here indicate that a more plausible pathophysiology for faulty facial

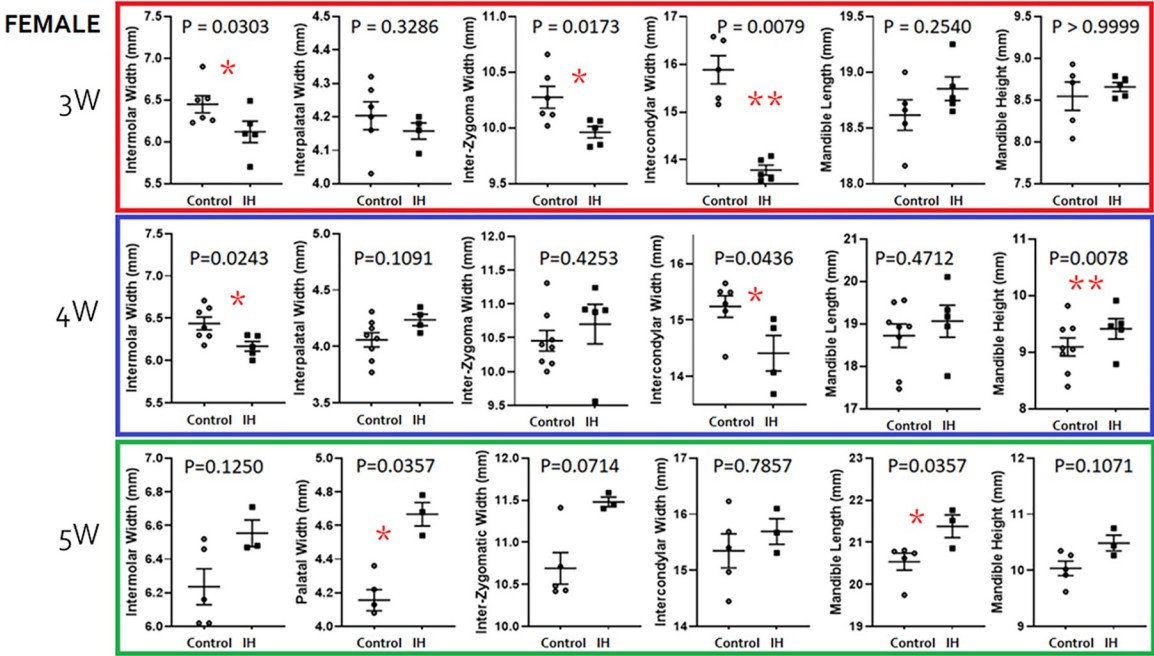

**Fig 8. Morphometric comparisons between the control *vs*. IH groups in 3- (red), 4- (blue) and 5- (green) week-old female pups.** Distribution-free statistical methods were used to test implications.

bone development is impaired breathing, leading to IH that triggers exaggerated sympathetic outflow at an early age.

Circulating NE levels provides an estimate of sympathetic discharge [20]. Our findings indicate that 'IH stress' elevates circulating NE levels, probably *via* an increased sympathetic transduction [40, 41]. Thus, elevated blood levels of NE most likely lead to defective bones *via* dysregulated adrenergic signaling in bones [17, 42, 43]. β2-adrenergic receptor-mediated signaling in response to the elevated NE inhibits bone formation and triggers RANKL-mediated osteoclastogenesis, thus increasing bone resorption. This increased NE-mediated signaling in bone could, in part, result from damaged Purkinje cells in the cerebellum after IH-insults leading to reduction of protective dampening by Purkinje neurons on deep cerebellar fastigial nuclei, allowing enhanced sympathetic outflow [44]. Or, more directly, the elevated renal sympathetic outflow after acute intermittent hypoxic insults to the neural circuits associated with chemoreceptor and baroreceptor control of renal nerve activity also influence NE levels [10, 45]. Our current results showed that male IH pups maintained higher NE levels. Yet, female pups appeared to recover from IH-damage sometime between 3 and 4 weeks. Although this sexual disparity requires further investigation, we confirmed that an early brief exposure to IH elevates blood NE levels in our rat model.

Recently, Haraldsdottir et al. [8, 9] published a series of studies showing that preterm individuals suffer from relatively elevated sympathetic tone until adolescence and young adulthood. Their studies support other previous reports showing that bone mineral density measured in adults born with low birth weight preterm is significantly lower [22, 23] in a total of 189 samples included in this Swedish cohort. Their low birthweight preterm cohort (n = 55) exhibited a significantly shorter height in males and females, but weight measurements in the preterm group did not differ from those of control subjects. These results parallel our current outcomes, *i.e.*, body weights did not differ between IH and control pups at 4 and 5 weeks

despite the elevated NE levels in males. The Swedish data demonstrate that the very low body weight preterm showed a positive correlation between fat mass, BMI (body mass index) and Dickkopf-1 measurements, an inhibitor of bone formation. Our, as well as others' results suggest that inter-relationships amongst body weight, bone density, NE levels in blood, as well as autonomic nerve function would be useful targets for assessment in patients with a history of preterm birth.

We expected our findings on weight changes after IH challenge would down-regulate body weight, particularly in males. However, 3-week-old male groups only showed the IH effect, *i.e.* lower body weight (Fig 4). This outcome may indicate that the decreased bone density in the IH group could recover as the pups grow, or body weight continuously increases due to increased body fat accumulation. A major reason for the significant weight loss would be a lack of bone mass in the IH-groups at 3 weeks, as shown in Fig 4; however, a catchup growth appeared to overcome the deficit in body weight as they aged. This conclusion is also supported by other findings: IH-treated animals showed increased bone marrow spaces with elevated osteoclastic activity (Fig 6). Fig 5 exhibited an increased TRAP activity in subcondylar areas in the tissues obtained from male pups. We previously found a similar outcome in other long bones [1]. We suspected increased sympathetic function; thus, elevated activity of tyrosine hydroxylase (TH) antibody staining sympathetic fibers (Fig 6) in other bones was assumed as well. Increased sympathetic nerve activity in the dental alveolar bone mediated by β2-adrenergic receptors was clearly demonstrated in a recent report [18]. This finding supports our hypothesis, but also make further studies mandatory as bone growth continues and subnormal bone quality is maintained.

Several clinical comparison studies on human cohorts demonstrated that moderately preterm (born in gestational week of 33–36), very preterm (born in gestational week of 28–32) and extremely preterm (born before gestational week of 28) groups show an invariably higher (by 20–30%) incidence of malocclusion traits compared to individuals born full-term [37, 46, 47]. Those traits include bilateral posterior crossbites, deep bites, more frequent tooth impaction, and anterior open bite. Each study noted preterm birth as a risk factor for malocclusions. However, none of the reports clearly provided an underlying mechanistic pathophysiology for the early emergence of malocclusions even though long-term manipulations of jaw bones are a daily practice for orthodontic tooth movement. Yet, the most frequently alluded predisposing factor for narrow maxillary transverse dimensions was iatrogenic effects by a prolonged intubation, such as catheters in association with medical issues in neonatal intensive care units [37]. No studies measured changes in transverse dimensions of the maxilla or the mandible in neonatal infants. Our current study observed a narrow palate in IH-treated pups without life-support equipment.

The current study examined morphometric differences in the facial bones at 3 time points. We primarily observed significant transverse defects in the maxilla *i.e.*, a lack of intermolar widths (control *vs.* IH; 6.47 (n = 5) *vs.* 5.98 (n = 5), p = 0.02 in males; 6.43 (n = 7) *vs.* 6.16 (n = 5), p = 0.02 in females). Because this set of observations is congruent with other reports in humans, our hypothesis is supported. Other anatomical structures, particularly condyle widths, were significantly shorter in the IH pups (15.39 *vs.* 14.40, p = 0.01 in males; 16.29 *vs.* 14.06, p = 0.0003 in females). This finding suggests that inter-condylar widths, in addition to palatal widths, should be evaluated in future clinical studies on human children with preterm birth history before commencing orthodontic treatment.

Differences in gender effects are found in other IH experimental conditions [48]. Our previous studies reported maternal IH effects on developing offspring in pregnant rats [49], finding that male offspring have more profound changes in blood pressure, cardiac function, myocardial lipid peroxidase content, and abdominal adiposity. We also observed that IH

challenges *in utero* damaged postnatal skeletal growth in males more than females. Moreover, a 1 h IH challenge on neonatal rat pups immediately after birth showed sex-dependent differences [1]. Recently, another group reported a clear gender disparity using recurrent hypoxia to pregnant rats mimicking sleep apnea in humans [50].

Some of these relationships appear to parallel our current findings; for instance, elevated NE levels after IH challenge were apparent in male pups and sustained until 5 weeks. Morphometric studies revealed that transverse width measurements, such as inter-molar width and mandible size (in width and height) were clearly smaller than those in IH-treated males in comparison to the IH-effects revealed in female pups.

## Conclusion

One-time IH-exposure for 1 h immediate after birth disturbs bone remodeling processes for at least the first 5 postnatal weeks. This disturbance likely results from increased sympathetic discharge from the IH challenge. We found increased blood NE levels in IH-exposed groups when males and females were 3 weeks old. IH-exposed male pups showed elevated NE levels at 4 and 5 weeks postnatally as well when compared to age-matched control pups. We conclude that one-time transient IH exposure can induce defects in bone quality and transverse size of the maxilla and the mandible, particularly in male rats. Unfortunately, we did not include a positive control group such as pups treated with an adrenergic agonist. However, we showed that the enhanced sympathetic drive from an IH challenge elicits transient metabolic changes, with significant effects on weight. Both the bone deficiencies and potential metabolic alterations were sex-specific; the processes underlying the sex contributions are unknown. We suspect that our current findings may underlie the high frequency of disharmonies in facial bones such as posterior crossbite malocclusions in human subjects who experienced preterm birth and the resulting etio-pathophysiology. Although the metabolic changes appeared to be transient, the potential for long-term effects of exaggerated sympathetic outflow on pancreatic and other endocrine processes cannot be ruled out.

## Supporting information

**S1 Data.**
(DOCX)

## Author Contributions

**Conceptualization:** Eung-Kwon Pae, Ronald M. Harper.

**Data curation:** Eung-Kwon Pae.

**Formal analysis:** Eung-Kwon Pae.

**Funding acquisition:** Eung-Kwon Pae.

**Investigation:** Eung-Kwon Pae.

**Methodology:** Eung-Kwon Pae.

**Project administration:** Eung-Kwon Pae.

**Resources:** Eung-Kwon Pae.

**Software:** Eung-Kwon Pae.

**Validation:** Eung-Kwon Pae.

**Visualization:** Eung-Kwon Pae.

**Writing – original draft:** Eung-Kwon Pae.

**Writing – review & editing:** Eung-Kwon Pae, Ronald M. Harper.

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
