## [Decision Letter · Decision Letter 0]

29 Mar 2023

PONE-D-23-05663Intermittent Hypoxia in Neonatal Rodents Affects Facial Bone GrowthPLOS ONE

Dear Dr. Pae ¶,

Thank you for submitting your manuscript to PLOS ONE. After careful consideration, we feel that it has merit but does not fully meet PLOS ONE’s publication criteria as it currently stands. Therefore, we invite you to submit a revised version of the manuscript that addresses the points raised during the review process.

We look forward to receiving your revised manuscript.

Kind regards,

Ewa Tomaszewska, DVM Ph.D

Academic Editor

PLOS ONE

Journal Requirements:

   "EP received Biomedical Research Award from the American Association of Orthodontists Foundation (AAOF) at https://www.aaofoundation.net/ for the study. PI was a sole investigator. This award supported the study fully."

   "This research was supported by AAOF grant awarded to E. Pae."

   "EP received Biomedical Research Award from the American Association of Orthodontists Foundation (AAOF) at https://www.aaofoundation.net/ for the study. PI was a sole investigator. This award supported the study fully."

Reviewers' comments:

Reviewer's Responses to Questions

**Comments to the Author**

1. Is the manuscript technically sound, and do the data support the conclusions?

Reviewer #1: Yes

2. Has the statistical analysis been performed appropriately and rigorously? 

Reviewer #1: Yes

3. Have the authors made all data underlying the findings in their manuscript fully available?

Reviewer #1: Yes

4. Is the manuscript presented in an intelligible fashion and written in standard English?

Reviewer #1: Yes

5. Review Comments to the Author

Reviewer #1: It is a very interesting and rigorous experiment. I did some suggestions. Please see below

Intermittent Hypoxia in Neonatal Rodents Affects Facial Bone Growth

Abstract: Without comments

Introduction: Without comments

Methods:

• How were selected the control and the experimental groups? Please explain it on the experimental design section. You mentioned age and sex matched on abstract, results and conclusions but not on methods section.

• Why all statistical inferences tests were two-tailed If your hypothesis assumed decreasing of mandibular bone quality and elevated norepinephrine levels. Then you assumed the direction of the difference. Please check.

Results: Without comments

Discussion:

• The following statement needs to be referenced: Although controversial, many previous malocclusion studies were based on the assumption that life-supporting equipment maintained in the oral cavity of preterm infants is the principal causal contributor to malocclusions

Limitation of the study: Please include on the discussion

Conclusion: without comments

Ethics Approval: It is presented

References: There are some of them with more than 10 years published, please consider the possibility to update

6. PLOS authors have the option to publish the peer review history of their article (what does this mean?). If published, this will include your full peer review and any attached files.

Reviewer #1: **Yes: **Angela Fernanda Espinosa Aranzales

---

## [Author Response · Author response to Decision Letter 0]

20 Apr 2023

The authors thank reviewers and their sincere recommendations.

---

## [Decision Letter · Decision Letter 1]

4 Jun 2023

Intermittent Hypoxia in Neonatal Rodents Affects Facial Bone Growth

PONE-D-23-05663R1

Dear Dr. Eung-Kwon Pae ,

We’re pleased to inform you that your manuscript has been judged scientifically suitable for publication and will be formally accepted for publication once it meets all outstanding technical requirements.

Kind regards,

Ewa Tomaszewska, DVM Ph.D

Academic Editor

PLOS ONE

Additional Editor Comments (optional):

Reviewers' comments:

Reviewer's Responses to Questions

**Comments to the Author**

1. If the authors have adequately addressed your comments raised in a previous round of review and you feel that this manuscript is now acceptable for publication, you may indicate that here to bypass the “Comments to the Author” section, enter your conflict of interest statement in the “Confidential to Editor” section, and submit your "Accept" recommendation.

Reviewer #1: All comments have been addressed

2. Is the manuscript technically sound, and do the data support the conclusions?

Reviewer #1: Yes

3. Has the statistical analysis been performed appropriately and rigorously? 

Reviewer #1: Yes

4. Have the authors made all data underlying the findings in their manuscript fully available?

Reviewer #1: Yes

5. Is the manuscript presented in an intelligible fashion and written in standard English?

Reviewer #1: Yes

6. Review Comments to the Author

Reviewer #1: The authors addressed all my comments, it is a very interesting experiment. I find now the manuscript more solid. Thank you

7. PLOS authors have the option to publish the peer review history of their article (what does this mean?). If published, this will include your full peer review and any attached files.

Reviewer #1: **Yes: **Angela Fernanda Espinosa Aranzales

---

## [Editor Report · Acceptance letter]

8 Jun 2023

PONE-D-23-05663R1 

Intermittent hypoxia in neonatal rodents affects facial bone growth 

Dear Dr. Pae:

I'm pleased to inform you that your manuscript has been deemed suitable for publication in PLOS ONE. Congratulations! Your manuscript is now with our production department. 

Kind regards, 

on behalf of

Professor Ewa Tomaszewska 

Academic Editor

PLOS ONE